# Niraparib and Advanced Ovarian Cancer: A Beacon in the Non-BRCA Mutated Setting

**DOI:** 10.3390/ph16091261

**Published:** 2023-09-06

**Authors:** Mauro Francesco Pio Maiorano, Brigida Anna Maiorano, Annalucia Biancofiore, Gennaro Cormio, Evaristo Maiello

**Affiliations:** 1Obstetrics and Gynecology Unit, Department of Biomedical Sciences and Human Oncology, University of Bari “Aldo Moro”, 70126 Bari, Italy; m.maiorano23@studenti.uniba.it; 2Oncology Unit, Casa Sollievo della Sofferenza IRCCS, 71013 San Giovanni Rotondo, Italy; e.maiello@operapadrepio.it; 3Pharmacy Unit, Casa Sollievo della Sofferenza IRCCS, 71013 San Giovanni Rotondo, Italy; a.biancofiore@operapadrepio.it; 4Gynecologic Oncology Unit, IRCCS Istituto Tumori “Giovanni Paolo II”, 70124 Bari, Italy; gennaro.cormio@uniba.it; 5Department of Interdisciplinary Medicine (DIM), University of Bari, 70126 Bari, Italy

**Keywords:** niraparib (ZEJULA^®^), PARP, ovarian cancer, BRCA, target therapy

## Abstract

Ovarian cancer (OC) is the eighth most common cancer among the female population and the most lethal of all the female reproductive system malignancies. Poly (ADP-ribose) polymerase inhibitors (PARPis) have reshaped the treatment scenario of metastatic OC in the maintenance setting post platinum-based chemotherapy. Niraparib is the first Food and Drug Administration (FDA)- and European Medical Agency (EMA)-approved PARPi as maintenance therapy for platinum-sensitive OC, regardless of BReast CAncer gene (BRCA) status, in first-line patients, with a recent restriction to germline BRCA mutations in second-line patients. In this review, we comprehensively summarized the pharmacological properties of niraparib, alongside the efficacy and safety data of the main trials leading to the current approvals, and discussed the future development of this agent.

## 1. Introduction

Ovarian cancer (OC) is the eighth most common female cancer worldwide, with an incidence of 8.1 cases/100,000 inhabitants/year [1]. Some 90% of OC are of epithelial cell type: among them, 75% of cases are represented by the high-grade serous OC (HG-SOC) and the remaining 10% are non-epithelial OC, which include mainly germ cell tumors, sex cord-stromal tumors, and extremely rare tumors such as small cell carcinomas [2]. Some 57% of OC cases are diagnosed as metastatic, with poor 5-year survival rates (median 30.8%). Indeed, OC bears the highest mortality rate among gynecological tumors, with 5.4 deaths/100,000 inhabitants/year [1]. Type I epithelial OC is suggested to be a relatively indolent and genetically stable group of tumors that typically arise from recognizable precursor lesions, such as endometriosis or borderline tumors with low malignant potential. In contrast, type II epithelial OC includes a more biologically aggressive group of tumors from the outset, with a propensity for metastasis even from small primary lesions: these include HG-SOC [3]. The first-choice treatments for advanced OC are platinum-based regimens, but after initial benefits, two out of three patients relapse mainly within the first two years [4,5,6,7,8,9]. Poly (ADP-ribose) polymerase (PARP) inhibitors (PARPis) are a class of antitumor agents whose mechanism of action relies on the exploitation of the defective DNA repair pathways in homologous recombination repair (HRR) gene deficient cells, a group of genes, including also breast cancer (BRCA)-1 and BRCA2, crucial for double-stranded breaks (DSBs) and interstrand crosslinks (ICLs) repairing pathways, whose mutations cause defective DNA repair, and finally lead to apoptosis [10,11]. When lacking HRR function, DSBs will be processed by alternative but more error-prone repair pathways, such as the non-homologous end-joining repair (NHEJ), which impairs cell survival and induces apoptosis [12]. Of note, 50% of OCs carry HRR deficiency (HRD), with 22% of cases bearing a germline or somatic mutation of BRCA1 and BRCA2, thus indicating the use of PARPis as a possible target therapy [13]. Indeed, PARPis are approved as maintenance after a platinum-based chemotherapeutic regimen. However, in first line patients, the approval of olaparib only for BRCA mutant patients as monotherapy, or HRD in association with bevacizumab, left an unmet need regarding the usefulness of PARPis for managing patients without genetic alterations [14].

With our review, we aim to summarize the pharmacological properties, principal efficacy, and safety data for the approved indications, as well as future therapeutic developments of niraparib (ZEJULA^®^). This might meet the need for new therapeutic options in the maintenance setting of the majority of advanced OC patients, showing its efficacy regardless of BRCA status in first line patients, and being the first PARPi approved with this indication, but recently restricted to BRCA mutant cases in patients after the first recurrence.

## 2. Pharmacodynamics and Pharmacokinetics of Niraparib

Niraparib is a tosylate monohydrate crystalline salt with a water solubility of 0.72 mg/mL. It is a 2-[4-(piperidin-3-yl)phenyl]-2H-indazole-7-carboxamide (C_19_H_20_N_4_O), with molecular mass 320.4 g/mol [15]. Niraparib acts as a highly selective, orally available inhibitor of the nuclear proteins PARP-1 and -2. Similarly to the other PARPis, the structure was created after the evidence that nicotinamide weakly inhibits PARP. Potency is weak due to the rotation of the amide bond. Unlike olaparib and rucaparib—which use an amide ring to restrain amide rotation—niraparib overcomes this problem by positioning a hydrogen bond-accepting group so that the NH anti-carbonyl amide forms an intracellular H-bond [16,17]. PARP-1 and -2 are nuclear proteins that contain a DNA-binding domain and a catalytic domain: this allows a conformational protein rearrangement that alters the catalytic domain, increasing its activity up to 500-fold [17,18].

Zejula (niraparib) is an orally available PARPi. In vitro, niraparib (MK-4827) inhibits the enzymatic activity of several PARP family members, but had greater than 500-fold potency against PARP-1 and PARP-2 (inhibition concentration of 50% [IC50], values of 2.8 and 0.6 nM, respectively) [16,17,18]. In vitro, niraparib inhibits PARP-1 and -2, which cause DNA damage, apoptosis, and cell death by increasing the formation of PARP–DNA complexes [18]. In vivo, niraparib was shown to reduce tumor growth in HG-SOC models, with increased tumor activity in HRD tumor models [19]. Niraparib inhibits dopamine, norepinephrine, and serotonin transporters by affecting heart rate and blood pressure [19,20].

Niraparib is rapidly absorbed, reaching maximum plasma concentration within 3–4 h. After that, the plasma concentration decreases in a biphasic pattern [14]. Niraparib has a bioavailability of 73% and binds to 83% of plasma proteins. The mean volume of distribution is 1220 L [15]. Niraparib is mainly metabolized by amide hydrolysis catalyzed by carboxyethyl-esterases, forming an inactive acid metabolite (M1) that subsequently undergoes glucuronidation [14]. 

Neither niraparib nor M1 is an inhibitor of any active substance-metabolising Cytochrome P450 (CYP) enzymes (CYP1A1/2, CYP2B6, CYP2C8, CYP2C9, CYP2C19, CYP2D6, and CYP3A4/5). Even though inhibition of CYP3A4 in the liver is not expected, the potential to inhibit CYP3A4 at the intestinal level has not been established at relevant niraparib concentrations. Therefore, caution is recommended when niraparib is combined with active substances the metabolism of which is CYP3A4-dependent, and those having a narrow therapeutic range. Neither niraparib nor M1 is a CYP3A4 inducer in vitro. In vitro, niraparib weakly induces CYP1A2 at high concentrations and the clinical relevance of this effect could not be completely ruled out. M1 is not a CYP1A2 inducer. Therefore, caution is recommended when niraparib is combined with active substances the metabolism of which is CYP1A2-dependent [18]. Moreover, niraparib weakly inhibits organic cation transporter-1 (OCT-1): caution is advised with concomitant drugs that are substrates of these transporters [17,19]. Niraparib shows a long terminal half-life (2 days) with a once-daily dosing regimen [15]. The principal elimination routes of niraparib and its metabolites are the hepatic/biliary and renal pathways. At the FDA-approved dose, 47.5% of the drug is excreted in the urine and 38.8% in the feces over 21 days [17,20,21].

### Dosage and Administration Route

Niraparib is administered per os at a dose of 300 mg once daily (OD) in the recurrent setting. A starting dose of 200 mg for patients weighing less than 58 kg may be considered. As first-line maintenance, the fix dose is 300 mg. However, if weight is ≤77 kg, and baseline platelet count ≤150,000/μL, 200 mg OD is the starting dose [17,21]. 

## 3. Niraparib Dosage and Use in Special Populations

### 3.1. Renal and Liver Impairment

As niraparib undergoes hepatic and renal elimination, its pharmacokinetic can be impacted in case of these organs’ impairment. According to a pharmacokinetic analysis, niraparib area-under-the-curve (AUC) inf was increased by 56% in patients with moderate hepatic impairment, compared with subjects with normal hepatic function [22]. Therefore, niraparib dose should be reduced to 200 mg OD in patients with moderate hepatic impairment [17,21,22]. Safety and pharmacokinetic data are lacking in patients with severe hepatic impairment; thus, in these patients, niraparib should be used cautiously [17,21].

Patients with mild to moderate renal impairment had minimal altered niraparib exposure if compared to subjects with normal renal function; thus, dose adjustment is not necessary. However, data are again lacking for patients with severe renal impairment or undergoing dialysis, and niraparib should be used cautiously in this population [23].

### 3.2. Old Patients

Although most OCs develop after the age of 65, barely 1 out of 3 patients is aged ≥65 in the major clinical trials of niraparib. Data regarding this population are available from the PRIMA/ENGOT-OV26/GOG-3012 trial in the newly diagnosed, and the ENGOT-OV-16/NOVA and NORA trials in platinum-sensitive, recurrent OC (PS-ROC) patients [24,25,26,27,28].

In these studies, ≥65 patients ranged from 13.9 to 39.4%. In a recent meta-analysis, we showed no differences in terms of efficacy between older and younger patients. Moreover, no increased risk of hematologic toxicity emerged in ≥65 women [29]. These data favor using full-dose niraparib in the older population. Nevertheless, trials specifically focusing on this age group should be conceived, considering the median age at diagnosis, and the aging population expected in the coming years.

## 4. Therapeutic Efficacy of Niraparib

### 4.1. Maintenance Treatment of Recurrent, Platinum-Sensitive, Advanced Ovarian Cancer

In March 2017, the Food and Drug Administration (FDA), and then the European Medical Agency (EMA), approved niraparib for the maintenance treatment of adult patients with recurrent epithelial ovarian, fallopian tube, or primary peritoneal cancer after a complete or partial response (CR or PR) to platinum-based chemotherapy [18,21]. 

The approval was based on the results of the ENGOT-OV16/NOVA trial (NCT01847274), a phase III study in which 553 patients with PS-ROC were randomized to receive niraparib or a placebo (PBO). In total, 203 patients had a germline BRCA mutation (gBRCAm), while 350 were gBRCA wild type (Non-gBRCA). Progression-free survival (PFS) was the primary endpoint. In the gBRCAm subgroup, patients reached a longer PFS with niraparib than PBO (21.0 vs. 5.5 months; hazard ratio [HR] 0.27). In the non-gBRCA subgroup, patients with PR or CR achieved a significant benefit with niraparib, reaching an mPFS of 9.3 versus 3.9 months (HR 0.45) [30]. 

Non-gBRCA patients benefitted from niraparib beyond the first progression, BRCAm patients reached an HR of 0.67. After a median follow-up (mFU) of 5.6 years (cut-off: October 2020), the non-gBRCA subgroup reached an adjusted median overall survival (mOS) of 31.03 months with niraparib compared with 35.9 months of PBO (HR 0.97). For gBRCAm patients, OS data showed an improved trend with niraparib (mPFS 43.8 vs. 34.1 months; HR 0.66) [31]. However, at the 2023 meeting of the Society of Gynecologic Oncology (SGO), the mFU was more than 75 months across groups (cut-off: March 2021). In the gBRCAm cohort, mOS was 40.9 months with niraparib and 38.1 months with PBO (HR 0.85; 95% CI, 0.61–1.20). In the non-BRCA cohort, mOS was 31.0 months with niraparib and 34.8 months with PBO (HR 1.06). Therefore, the FDA, but not the EMA, restricted niraparib indication to only patients with gBRCA mutations [18,32]. 

### 4.2. First-Line Monotherapy Maintenance Treatment of Advanced Platinum-Sensitive Ovarian Cancer

In April 2020, the FDA approved niraparib for the first-line maintenance treatment of patients with advanced epithelial ovarian, fallopian tube, or primary peritoneal cancer with CR or PR to first-line platinum-based chemotherapy. Niraparib is the first and only PARPi approved with this indication, regardless of BRCA or HR status [21].

The approval followed the PRIMA/ENGOT-OV26/GOG-3012 (NCT02655016), a phase III randomized trial assessing the efficacy of niraparib 200 mg OD versus PBO. A total of 733 patients were involved, of which 373 (50.9%) were HRD. In this group, mPFS was significantly longer in the niraparib than in the PBO arm, reaching 21.9 versus 10.4 months, respectively. In the overall population, mPFS was 13.8 versus 8.2 months. OS data were immature; however, the rate of OS at 24 months was 84% in the niraparib and 77% in the PBO group. As shown by the PRIMA trial, those who received niraparib had significantly longer PFS than those who received PBO, regardless of the presence or absence of HRD [24].

Table 1 summarizes the main trials employing niraparib as maintenance treatment for advanced OC.

## 5. Tolerability of Niraparib

Examining the tolerability of niraparib as maintenance therapy in advanced OC, we found that almost every patient experienced at least one adverse event (AE) of any grade, ranging from 98.8% to 100% of patients receiving niraparib versus 68.9% to 91.8% in the PBO arm. Hematologic toxicities were the most commonly reported all-grade AEs. In the NOVA trial, 50.1% patients developed anemia, 30.2% neutropenia, and 61.3% thrombocytopenia (TCP). In the PRIMA study, anemia occurred in 63.4% of patients, neutropenia in 26.4%, TCP in 45.9%. Hematological toxicities are common PARPis class AEs, representing the leading cause of dose reduction and treatment discontinuation. They usually occur early after treatment start and recover after a few months, and seem more frequent after niraparib than olaparib. Anemia is the most common, and it might be related to PARP2 inhibition that affects the differentiation of erythroid progenitors, reducing erythrocyte life expectancy, even with increased erythropoietin plasma concentrations. A recent study found that baseline body weight and platelet counts might be identified as predictors of dose modification in patients treated with niraparib at 300 mg OD [33]. Niraparib-associated TCP might be related to a reversible decrease in megakaryocyte proliferation and maturation [33,34]. Patients with a baseline bodyweight ≤ 77 kg, or platelets ≤ 150,000/μL, experienced more ≥G3 TCP during the first month of treatment, suggesting that these patients might benefit from a reduced starting dose of niraparib (200 mg OD) [35]. The PRIMA study was amended to individualize the niraparib starting dose based on body weight (cut-off: 77 kg) and/or platelet count (cut-off: 150,000/μL). The results of rational adjustment of dosage to reduce adverse reactions (RADAR) presented at ESMO 2018 showed that 159 patients receiving an individualized dose based on body weight and platelet count had lower ≥G3 AEs compared to 471 patients starting with niraparib 300 mg [35]. The recommended starting dose of niraparib is 200 mg (two 100-mg capsules) OD. However, for those patients who weigh ≥ 77 kg and have baseline platelet count ≥150,000/μL, the recommended starting dose of Zejula is 300 mg (three 100-mg capsules) OD [18].

As hematological toxicities are more frequent among patients using niraparib, especially TCP, differently from the general PARPis treatment suggestion of a monthly complete blood count, the FDA recommends weekly testing in the first month for patients starting niraparib [21,36]. Transfusions are generally recommended for symptomatic anemia and hemoglobin values ≤ 7 g/dL. A bone marrow analysis is recommended in case of severe hematologic AEs lasting over 4 months. In fact, another rare but severe PARPis class effect is the onset of myelodysplastic syndrome (MDS) and acute myeloid leukemia (AML), usually after long-term treatment. Among 1785 patients treated with niraparib in clinical trials, MDS/AML occurred in 15 cases (0.8%) [17,21,24,30].

Among PARPis, niraparib is the most associated with cardiovascular toxicity, including palpitations, tachycardia, and hypertension. Hypertension of any grade was reported by 19.3% of patients in the NOVA trial [24,30]. The mechanisms behind niraparib cardiovascular toxicity are not well known. In fact, previous reports indicated that PARP1 activation might be associated with hypertension and myocardial dysfunction, thus suggesting a cardioprotective effect of PARPis [37,38]. The cardiovascular toxicity of niraparib might be related to a disruption of dopamine, norepinephrine, and serotonin metabolism, as niraparib can inhibit their cellular uptake. Thus, close blood pressure and heart rate monitoring should occur during niraparib treatment, at least weekly for the first two months, then monthly, especially for patients with baseline hypertension and cardiovascular disorders. Hypertension should be managed using antihypertensive drugs or dose modification, if necessary [17,21,36].

Niraparib is also commonly associated with gastrointestinal AEs, such as nausea or vomiting. In the NOVA trial, 73.6% of patients developed nausea of any grade, and 34.3% vomiting. In the PRIMA study, nausea occurred in 57.4% of cases, vomiting in 22.3%. [24,30,32]. Prokinetic and antihistamine drugs can be administered daily, and persistent nausea or vomiting can be managed with antiemetic drugs, such as metoclopramide, prochlorperazine, phenothiazine, dexamethasone, olanzapine, haloperidol, or lorazepam. 

Fatigue is also a common class effect of PARPis. In total, 59.4% and 34.7% of patients developed fatigue in the NOVA and PRIMA trials, respectively. Patients can be managed using non-pharmacological approaches first, but psychostimulants such as methylphenidate and ginseng are currently being investigated [36] (Figure 1).

Regarding severe toxicity, ≥G3 AEs were reported by 56–75.5% of patients treated with niraparib versus 6.6–22.9% of patients receiving PBO: among these, hematological toxicities were by far the most frequently experienced. Anemia occurred in 25.3% of patients in the NOVA trial, and 31.0% in the PRIMA study. Neutropenia presented in 19.6% of patients in the NOVA trial, 12.8% in the PRIMA study. TCP was reported by 33.8% of patients in the NOVA trial, and 28.7% in the PRIMA. Hypertension was reported by 8.2% of patients in the NOVA trial, and 6% in the PRIMA study. Nausea and vomiting occurred, respectively, in 3.0% and 1.9% of patients in the NOVA, and 1.2% and 0.8% in the PRIMA trial. Fatigue presented in 8.2% of patients in the NOVA study, and 1.9% in the PRIMA trial (Figure 2). 

In the NOVA trial, the rate of dose reduction (DR) was 66.5% in the niraparib arm versus 14.5% in the PBO arm, while the dose interruption (DI) rate was 68.9% versus 5%. Treatment discontinuation occurred in 14.7% of patients in the niraparib vs. 2.2% in the PBO group [30]. 

In the PRIMA trial, haematologic AEs were the foremost responsible for DR (70.9% vs. 82%), DI (19.5% vs. 2.5%), and treatment discontinuation (12% vs. 2.5%) [24]. 

Another supposed risk of OC patients treated with PARPis is the development of new secondary primary malignancies (SPMs), whose actual incidence is difficult to estimate, as almost all patients treated with PARPis also receive other DNA-damaging drugs, such as platinum derivatives. However, the risk of developing SPMs, such as breast, thyroid, and rectal cancer, was not found to be increased after niraparib (0.9%) compared with PBO (0.7%; *p* = 0.62) in a recent meta-analysis of 23 randomized clinical trials, including 8857 patients, thus suggesting no additional close monitoring is needed during PARPis [24,30,39]. Additionally, several animal studies show that niraparib is embryo-toxic and teratogenic. Human studies are limited. However, due to its mechanism of action, niraparib could damage embryo and fetus. For example, PARP1 upregulation is fundamental for embryo implanting. Therefore, fertile women should avoid niraparib during pregnancy and until at least six months after birth. Breastfeeding is also contraindicated during and until one month after the last dose of niraparib [17,21,40].

## 6. Future Perspectives and Conclusions

PARPis are reshaping the OC therapeutic scenario, with niraparib being a forerunner for this process, especially in the non-BRCA/HRR mutated setting. So far, niraparib remains the only approved PARPi in the PS-ROC maintenance setting, regardless of BRCA status, even if the most recent updated from the studies of recurrent patients are questioning this paradigm. Moreover, niraparib is effective also in BRCA-mutated patients. Recently, the phase III PRIME trial demonstrated the efficacy of niraparib versus PBO in newly diagnosed Chinese OC patients after first-line chemotherapy, including those who were resected at primary debulking surgery with an individualized starting dose of 300 mg OD, reduced to 200 mg OD in case of body weight ≤77 kg and/or platelet count ≤150,000/μL. A total of 384 patients were randomized and, after an mFU of 27.5 months, mPFS was 24.8 vs. 8.3 months (HR 0.45, 95% CI 0.34–0.60; *p* < 0.001). The study enlarges the audience of candidates of niraparib, as it includes some categories originally excluded from the PRIMA, such as the resected stage III [41]. 

Despite a slightly higher incidence of haematologic and cardiovascular toxicities compared to other PARPis, niraparib maintains a generally good safety profile and quality of life. A peculiar advantage of niraparib—compared with other PARPis such as olaparib—is the single daily dosage, with the possibility of dose personalization based on patients’ clinical and laboratory characteristics, and limited pharmacological interactions.

Several questions, however, remain to be answered, especially regarding the combination with other agents, the platinum-resistant (PR)-ROC setting, and the treatment after progression to PARPis. 

Observing that, through DNA damage, PARPis stimulate neo-antigen production, therefore augmenting the tumor mutational burden, a huge therapeutic combination of PARPis is with immune-checkpoint inhibitors (ICIs). PD-L1 expression, in fact, is upregulated by PARPis, as they are able to switch the tumor microenvironment towards a higher immune-responsiveness, as well as increasing tumor-infiltrating lymphocytes. In addition, PARPis seem to activate the STING pathway that stimulates interferon-γ dependent immune cells [42]. Clinical models have also demonstrated that PARP inhibition inactivates glycogen synthase kinase 3 (GSK3) and upregulate PD-L1 in a dose-dependent manner, consequently suppressing T-cell activation, and resulting in enhanced cancer cell apoptosis [43]. In the phase I-II TOPACIO/Keynote-162 trial (NCT02657889), 62 women with PR-ROC were enrolled to receive niraparib plus pembrolizumab. Most patients were BRCAwt (79%) or HRP (53%): they reached an ORR of 25% and a DCR of 68% [44]. Anti-angiogenic compounds should be also considered to improve the effectiveness of PARPis and ICIs, with a more extensive molecular and genetic characterization of OC that could be very useful for the assessment of the treatment response and comprehensive understanding of adaptive mechanisms. Moreover, considering that the vascular endothelial growth factor (VEGF) pathway interacts with the phosphoinositide 3-kinase (PI3K)/protein kinase B (AKT)/mammalian target of rapamycin (mTOR) pathway, within the context of precision medicine, mTOR- and mitogen-activated protein kinase (MEK)-inhibitors might also be considered, according to the specific mutational profile [45]. As PARP inhibition decreases angiogenesis, and hypoxia and VEGF-receptor 3 (VEGFR3) inhibition also induces the down-regulation of HR proteins, giving a rationale for the combination of PARPis and anti-angiogenetic drugs [45,46,47,48]. The NSGO-AVANOVA2/ENGOT-ov24 phase II trial compared niraparib plus bevacizumab versus niraparib alone in a cohort of 97 patients with PS-ROC in a chemo-free setting. Niraparib plus bevacizumab significantly improved PFS (11.9 vs. 5.5 months; *p* < 0.0001). ≥G3 AEs were reported in 65% of patients who received niraparib plus bevacizumab, and 45% in the niraparib monotherapy, most commonly anemia, and TCP. Hypertension and proteinuria raised in the group receiving olaparib plus bevacizumab [49]. In fact, as combination trials try to meet the need for new therapeutic options, concerns may be expressed about potentially new and/or augmented AEs. A total of 105 patients with PS-ROC, also previously treated with bevacizumab, were enrolled in the phase II OVARIO trial to receive the combination of niraparib and bevacizumab. mPFS was 19.6 months in the overall population, and 28.3 and 14.2 months in the HRD and HRP subgroups, respectively. No new safety concerns emerged in this study [50]. The ongoing phase II (KGOG 3056)/NIRVANA-R trial will assess the efficacy of niraparib plus bevacizumab in PS-ROC patients already treated with a PARPi [51].

Effective treatment options are currently limited in the PR-ROC setting, and patients relapsing to platinum-based regimens within 12 months usually exhibit poorer responses to subsequent lines of treatment. Using anti-angiogenetics improves PFS, but not without concerns about AEs [52]. Trials involving PR-ROC patients have not yet resulted in benefits in terms of survival or response rates, thus justifying further research and clinical trials with novel agents [53]. The phase II ANNIE study (NCT04376073) analyzed the efficacy and safety of the combination niraparib plus the tyrosine kinase inhibitor (TKI) anlotinib. Among the 40 patients enrolled, ORR was 50%, reaching 100% in gBRCAm patients. As no new safety concerns emerged, this chemo-free combination might represent a promising approach for PR-ROC patients [52]. 

Future research should explore the feasibility and efficacy of rechallenging niraparib after progression to a previous PARPi [53]. There is a strong need to identify patients that could indeed benefit from this treatment opportunity. In the phase IIIb OReO/ENGOT Ov-38 (NCT03106987) trial, there was a slight but statistically significant benefit when rechallenging olaparib after PARPi progression in heavily pre-treated patients (mPFS 4.3 vs. 2.8 months in BRCAm, 5.3 vs. 2.8 in HRD, 5.4 vs. 2.8 in HRP) [54]. Post-progression outcomes of PARPis studies are also evidencing the occurrence of platinum resistance in case of PARPis progression: these data should be further elucidated in order to develop treatment strategies [55,56]. Technologies of proteomics, such as mass spectrometry and protein array analysis, are advancing the dissection of the underlying molecular signaling events and the proteomic characterization of OC. Proteomic study of OC subtypes, as well as their adaptive responses to therapy, can uncover new therapeutic choices, which can reduce the emergence of drug resistance and potentially improve patient outcomes [57].

## Figures and Tables

**Figure 1 pharmaceuticals-16-01261-f001:**
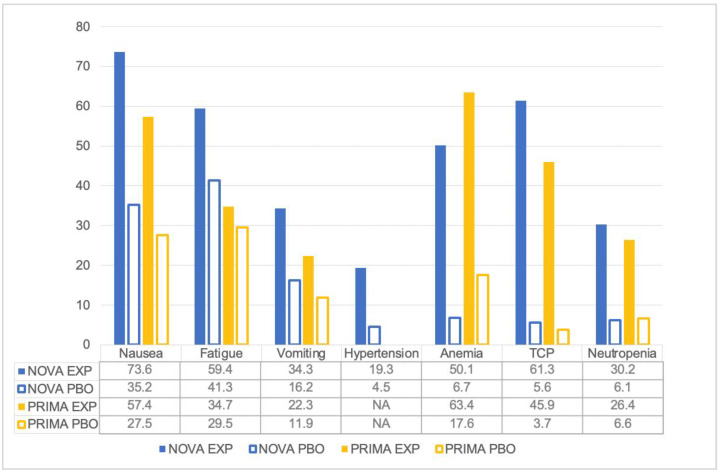
Percentage of patients experiencing all-grade AEs.

**Figure 2 pharmaceuticals-16-01261-f002:**
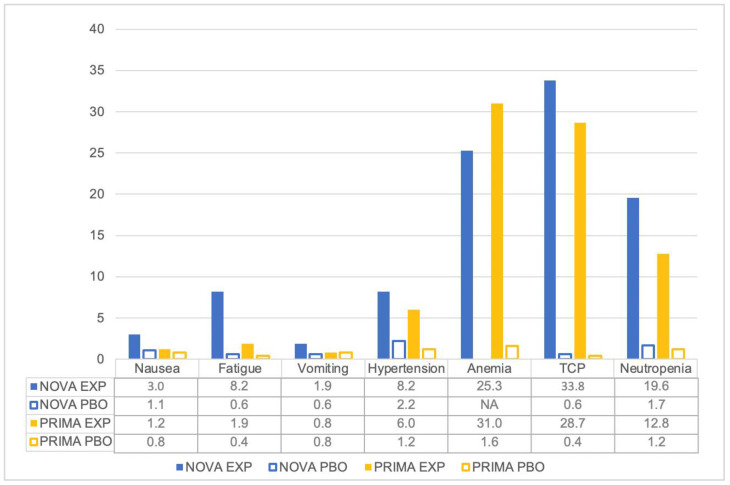
Percentage of patients experiencing ≥G3 AEs.

**Table 1 pharmaceuticals-16-01261-t001:** Summary of studies employing niraparib as maintenance in advanced OC.

Study Name (NCT)—Year	Phase	Target Population (*Number of pts)*	Niraparib Administration Protocol	Primary EP	Results	
					mPFS	mOS
Maintenance treatment of PS-ROC						
ENGOT-OV16/NOVA (NCT01847274)—2017	III	PS-ROC (n = 553) niraparib arm (n = 367) PBO arm (n = 179) gBRCAm (n = 203) Non-gBRCA (n = 350)	300 mg OD	mPFS	*gBRCAm subgroup*: 21.0 mos vs. 5.5 mos (HR 0.27; 95% CI 0.2–0.4; *p* < 0.0001) *Non-gBRCA subgroup*: 9.3 mos vs. 3.9 mos (HR 0.45; 95% CI, 0.34 to 0.61; *p* < 0.0001)	*gBRCAm subgroup*: 40.9 vs. 38.1 mos (HR 0.85, 95% CI 0.61–1.20) *Non-gBRCA subgroup*: 31.0 vs. 34.8 mos (HR 1.06, 95% CI 0.81–1.37)
First-line maintenance in newly diagnosed platinum-sensitive OC						
PRIMA/ENGOT-OV26/GOG-3012 (NCT02655016)—2020	III	Overall (n = 733) HRD+ (n = 373) niraparib arm (n = 487) PBO (n = 246)	300 mg OD (initial protocol) 200 mg OD (from Nov 2017)	mPFS	*Overall:* mPFS 13.8 mos vs. 8.2 mos (HR 0.62; 95% CI, 0.50–0.76, *p* < 0.001) *HRD+:* mPFS 21.9 mos vs. 10.4 mos (HR 0.43; 95% CI, 0.31–0.59, *p* < 0.001)	*Overall*: 24-mos OS rate 84% vs. 77% (HR 0.70; 95% CI, 0.44–1.11).

BRCA, breast cancer gene; CI, confidence interval; CHT, chemotherapy; CR, complete response; DoR, duration of response; EP, endpoint; gBRCA, germline BRCA mutation; non-gBRCA, BRCA wild-type; HR, hazard ratio; HRD, homologous recombination deficiency; HRDu, HRD status unknown; NA, not available; OC, ovarian cancer; OD, once daily; ORR, overall response rate; OS, overall survival; PBO, placebo; PFS, progression-free survival; PR, partial response; PS-ROC, platinum-sensitive recurrent ovarian cancer; pts, patients.

## Data Availability

Data sharing is not applicable.

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
