# Peer review of "Niraparib and Advanced Ovarian Cancer: A Beacon in the Non-BRCA Mutated Setting"

_pharmaceuticals, 2023, doi:10.3390/ph16091261_

Round 1
Reviewer 1 Report
In this paper, the Authors have reviewed the efficacy of utilization of poly(ADP-ribose) polymerase inhibitors (PARPis) to aid in the ovarian cancer (OC) treatment. The Authors have assessed recent progress in the application of the first FDA- and EMA-approved PARPi, niraparib. The review is focused on the pharmacological properties of niraparib. The scope of the review is broadened by considering also therapeutic efficacy and tolerability of niraparib. The Authors also discuss the challenges of effective treatment. This review will be invaluable for doctors and scientists involved in cancer therapy and the development of safe and effective cancer treatments. I recommend the paper for publication after minor revision addressing the issues listed below.
1. According to the National Library of Medicine PubChem CID 24958200, the chemical composition of niraparib compound is: 2-[4-(piperidin-3-yl)phenyl]-2H-indazole-7-carboxamide, with molecular formula C19H20N4O and its molar mass is 320.4 g/mol. Any derivative of this main compound, used in the experiments, should be emphasized, including the properties of the derivative (the molecular formula provided for Niraparib in the manuscript in line 64 is C26H30N4O5S and its molar mass 510.61 Da, suggesting a larger molecule than that indicated in the above PubChem listing). Please provide detailed information concerning the niraparib discussed.
2. The unit of concentration of niraparib against PARP-2 Mm is incorrect (line 75). Please correct appropriately.
3. In Figure 2 and 3 the variables should be added.
4. The explanation of abbreviations should be added at their first use in Abstract and the main text.
Author Response
We are extremely grateful to the reviewer for the comments and suggestions.
We modified the information regarding the chemical composition of niraparib.
We modified the measuring unit for concentration, which was incorrect.
We updated the two figures.
We added the missing abbreviations in the manuscript and the abstract.
Reviewer 2 Report
The title of the article reflects the content of the article.
In the "Abstract" section, the authors presented a brief summary of their review article. The purpose of this study was to review the pharmacological properties, data on the efficacy and safety of the main trials of niraparib. Niraparib is the first drug approved by the FDA and EMA PARPi (Poly (ADP-ribose) polymerase inhibitors) as a maintenance therapy for platinum-sensitive ovarian cancer. In addition, the authors discussed the future development of niraparib.
The "Keywords" presented in the article correspond to the content of the article and are necessary.
This review article is divided into 6 parts: «1. Introduction», «2. Pharmacodynamics and pharmacokinetics of niraparib», «3. Niraparib dosage and use in special populations», «4. Therapeutic efficacy of niraparib», «5. Tolerability of niraparib», «6. Future perspectives and conclusions». All sections are important and necessary.
In the section "1. Introduction", the authors presented the epidemiology, etiology of the disease, described the clinical course of ovarian cancer. The section lists platinum preparations as a class of antitumor agents for the treatment of ovarian cancer, describes the mechanism of action of platinum preparations. The authors indicated that poly (ADP-ribose) polymerase (PARP) inhibitors (PARPis) are used to combat relapses in patients with ovarian cancer. This class of antitumor agents is approved as a maintenance agent after a platinum-based chemotherapy course.
The aim of the study is to summarize the pharmacological properties, basic data on efficacy and safety, as well as to evaluate future therapeutic developments of Niraparib (ZEJULA®).
Further, the authors consistently provided information on the pharmacodynamics and pharmacokinetics, dosage and method of administration of niraparib. It is important that the authors indicated the use of the drug with caution in patients with severe hepatic or renal insufficiency. On the other hand, possible complications in older patients should be taken into account when prescribing niraparib. Sections 4 and 5 are the main sections of the article, where the authors, using a significant number of publications, evaluated the therapeutic efficacy of niraparib and the tolerability of niraparib as a maintenance therapy for ovarian cancer of various clinical course. The authors accompanied the sections «4. Therapeutic efficacy of niraparib» and «5. Tolerability of niraparib» with a table and figures. The presented table and figures are necessary and understandable.
In the final section, the authors summarized their study of niraparib. The authors rightly point out that niraparib retains a generally good safety profile and quality of life. The advantage of niraparib compared to other PARPI drugs is a single daily dose with the possibility of dose personalization depending on the clinical and laboratory characteristics of patients and limited pharmacological interactions. The authors а number of issues that should be discussed. For example, the possibility of using niraparib in combination with other drugs. The authors associate future studies with the need to study the feasibility and effectiveness of re-prescribing niraparib after switching to the previous PARPi therapy.
The article is interesting, timely and important for clinical medicine. The text of the article is written clearly. The manuscript did not cause any ethical problems. All links to publications in the "References" section are necessary and correct, made in the right style. Of the 56 references that are presented in the article, 38 references are from the last 5 years. I have no concerns about the similarity of this article with other articles published by the same authors.
Competing interests of authors do not create bias in the presentation of results and conclusions.
Author Response
We are extremely grateful to the reviewer for the comments and the comprehensive analysis of our manuscript.
Reviewer 3 Report
The review focuses on the Niraparib on the treatment of OC and delineates the different perspectives of it.
Major issues:
1. What is the point of discussion of so many details of pharmacodynamics and pharmacokinetics at the beginning of the paper ? what are they related to the
2. What is the main innovative point for this review ?
3. Compared to other PARPis, what is the main advantage of using Niraparib in the treatment of OC ?
4 You should also include the perspectives on the utility of PARPis in pregnancy women.
Minor issues:
1. "ad" in 2.1 subtitle may be a typo
2. It will be better and more straightforward if you can use more visualization method to represent data instead of just talking about it in the paragraph.
Author Response
We are very grateful to the reviewer for the interesting comments and suggestions. We modified the manuscript as indicated:
Regarding the major points, we modified the manuscript to include the most updated approvals of niraparib, specifically addressing the topic with the drug's clinical, pharmacological, and management elements for the readers. In the manuscript, we added more comments and elements to differentiate niraparib from other PARPis, and also regarding data of pregnant women.
Regarding the minor points, we added more graphical elements in Table 1 resuming the study characteristics, and reduced the amount of data in the text, both for efficacy but especially for toxicity.
Round 2
Reviewer 3 Report
I think major issues have been resolved.
Author Response
We would like to thank the reviewer, and also updated the manuscript with the editor's comments.